# Lipid Peroxidation Levels in Saliva and Plasma of Patients Suffering from Periodontitis

**DOI:** 10.3390/jcm11133617

**Published:** 2022-06-23

**Authors:** Tanja Veljovic, Milanko Djuric, Jelena Mirnic, Ivana Gusic, Aleksandra Maletin, Bojana Ramic, Isidora Neskovic, Karolina Vukoje, Snezana Brkic

**Affiliations:** 1Department of Dental Medicine, Faculty of Medicine, University of Novi Sad, 21000 Novi Sad, Serbia; milanko.djuric@mf.uns.ac.rs (M.D.); jelena.mirnic@mf.uns.ac.rs (J.M.); ivana.gusic@mf.uns.ac.rs (I.G.); aleksandra.maletin@mf.uns.ac.rs (A.M.); bojana.ramic@mf.uns.ac.rs (B.R.); isidora.neskovic@mf.uns.ac.rs (I.N.); karolina.vukoje@mf.uns.ac.rs (K.V.); snezana.brkic@mf.uns.ac.rs (S.B.); 2Dentistry Clinic of Vojvodina, 21000 Novi Sad, Serbia; 3Clinic for Infectious Diseases, Clinical Centre of Vojvodina, 21000 Novi Sad, Serbia

**Keywords:** malondialdehyde, oxidative stress, periodontitis, plasma

## Abstract

Lipid peroxidation (LPO) participates in the development of various diseases, including periodontitis, and malondialdehyde (MDA) is its terminal product. Therefore, in the present study, salivary and plasma MDA levels in 30 periodontitis patients were compared to those in 20 healthy controls, as well as in relation to periodontal therapy in order to assess its effectiveness. Periodontal status was assessed via plaque index, gingival index, papilla bleeding index, probing depth and clinical attachment level, while salivary and plasma MDA levels were determined by the ELISA method. The periodontitis group had a significantly greater salivary (2.99 pmol/µL) and plasma (0.50 pmol/µL) MDA levels relative to the healthy controls (1.33 pmol/µL and 0.40 pmol/µL, respectively). Three months after the periodontal therapy completion, although salivary MDA levels were significantly lower than those measured at the baseline (*p* < 0.001), the reduction in plasma MDA was not statistically significant (*p* > 0.05). These findings indicate that, while inflammatory processes in periodontium may increase local and systemic lipid peroxidation, periodontal therapy can result in a significant decrease in salivary, but not plasma, MDA levels.

## 1. Introduction

Periodontitis is an inflammatory disease of tooth-supporting structures, which leads to tissue destruction and tooth loss as a result of the interaction of dental plaque microorganisms and the host’s immune response. According to the World Health Organization (WHO) data, the global incidence of a more severe form of periodontal disease is estimated at 10–15% [1]. Therefore, this disease is a serious medical, economic and social problem. The significance of periodontitis is increased further due to its potential impact on systemic health. Although the exact mechanism of this relationship has not been fully elucidated, in recent years, increasing importance has been attached to oxidative stress arising in the course of periodontal disease as a potential risk factor in the development of some systemic diseases [2,3].

Oxidative stress arises due to an imbalance between the reactive oxygen species (ROS) and the antioxidant defense system, which results in damage to important cellular macromolecules, such as lipids, proteins and DNA. Lipid peroxidation (LPO) is the process of oxidative lipid damage. Due to the significant presence of lipids in the cell membrane and its subcellular organelles, they are the site of peroxidation onset. The LPO outcome is a marked change in the membrane permeability, which contributes to the degradation of cellular metabolism and homeostasis and may ultimately result in cell death [4].

ROS are unstable molecules with a short half-life, making them difficult to detect. This issue is overcome by using the terminal products of macromolecule oxidative damage, one of which is malondialdehyde (MDA), as it results from lipid oxidative damage. An extensive body of empirical data indicates that the MDA level in bodily fluids may be a reliable indicator of the extent of oxidative damage to cells in the body [5]. Available research indicates that MDA levels are increased due to cancer [6], atherosclerosis [7], diabetes [8], liver disease [9] and preeclampsia [10], as well as in smokers, while recent data suggest a strong link with periodontal disease [11,12,13,14,15].

Previously, the LPO level in saliva was believed to be predominantly affected by the extent of this process in blood. However, recent evidence indicates that locally induced oxidative stress plays a more important role. Namely, in the course of periodontal disease, free radical production by polymorphonuclear leukocytes increases as a defense mechanism against periodontopathogens. Existing studies indicate that, in the course of periodontal disease, LPO products such as MDA are released and participate in the progress of periodontal tissue inflammation and destruction [13,14,15,16,17,18]. However, periodontal therapy is believed to induce a reduction in this marker.

Furthermore, available research findings suggest that LPO products diffuse from the initial inflammation site and can be registered in the bloodstream [19]. This process could result in the emergence of certain systemic diseases. Tests conducted on laboratory animals have shown that experimentally induced periodontal disease can cause oxidative damage to the liver, thus increasing oxidative stress in the blood [20,21,22,23]. In addition, oxidative stress due to damage to periodontal tissues is posited to be involved in the development of atherosclerosis in laboratory animals [24]. As the number of such studies involving humans is limited, further research is required in order to reach more definitive conclusions about the impact of periodontal disease on systemic oxidative stress.

The objectives of the present investigation were to: (1) compare the level of LPO in plasma and saliva of patients with and without periodontal disease; (2) examine the impact of periodontal therapy on the salivary and blood LPO levels in patients with periodontitis; and (3) examine the link between the level of LPO and clinical markers of periodontal status.

The overall aim of the study was to determine if periodontitis can result in an increase in local and systemic oxidative stress levels, which would potentially provide a link between periodontitis and systemic diseases.

## 2. Materials and Methods

### 2.1. Subjects

This research involved 50 patients. All participants were informed in writing about the study aims, the nature of their involvement and the intended use of the results obtained, after which they signed the consent form. The study was approved by the local ethics committee. All procedures performed in the study involving human participants were in accordance with the Declaration of Helsinki.

The study inclusion criteria were being 30–70 years old, having at least 20 teeth and being systemically healthy. Patients were excluded from the study if they met any of the following criteria: periodontal therapy in the previous six months, use of antibiotics in the last three months, use of any vitamin supplementation and pregnancy.

### 2.2. Periodontal Examination

In all study participants, periodontal status was assessed via plaque index (PI) [25], gingival index (GI) [26], papilla bleeding index (PIB) [27], probing depth (PD) and clinical attachment level (CAL). We utilized the same indices in our earlier studies, as empirical evidence indicated that they are most representative of periodontium conditions [28,29,30]. Measurements were performed on mesio-buccal, disto-buccal, mid-buccal and mid-lingual tooth surfaces using Michigan ‘O’ probe with William’s markings. All measurements were performed by the same periodontist.

### 2.3. Study Groups

Criteria for inclusion in the group of patients with periodontal disease were as follows: at least two sites per quadrant with PD ≥ 4 mm, 30% bone loss and gingival inflammation [31]. Thirty periodontitis patients formed the experimental group, designated as Group A. The control group (Group B) consisted of 20 patients with no signs of damage to periodontal supporting structures. Before commencing the study, we used power analysis, which indicated that 18 participants would be sufficient for achieving a 95% power and 95% significance level. As our study design involved pre- and post-treatment evaluations, to account for possible attrition, we included 20 and 30 patients in the control and the experimental group, respectively.

In order to assess the influence of the GI, PD and CAL on the MDA levels, periodontitis patients were divided into six subgroups. The GI levels allowed a further division into two subgroups, comprising of individuals with moderate (Loë–Silness 0.1–2) and severe (Loë–Silness 2.1–3) inflammation. Similarly, two subgroups were formed based on the PD, with patients with PD ≥ 5 mm on more than 20% of sites forming one subgroup and those with PD ≥ 5 mm on fewer than 20% of sites forming the other. Finally, CAL was used to separate patients into a subgroup with the mean value of CAL ≥ 3 mm and CAL < 3 mm, respectively.

### 2.4. Sample Collection and Preparation

Determination of oxidative stress markers was conducted in mixed unstimulated saliva samples taken in the morning from patients who were instructed not to drink or eat prior to attending the appointment. Salivary samples were centrifuged at 3000× *g* for 10 min at room temperature, after which time the supernatant was isolated and stored at −80 °C until required for analysis.

Blood samples were taken from a fingertip and collected in special tubes coated with EDTA (Kabe Labotechnik, Nümbrecht-Elsenroth, Germany) and were transported to the laboratory, where they were immediately centrifuged at 3000× *g* for 10 min. The thus obtained plasma samples were stored at −80 °C until required for analysis.

### 2.5. MDA Assay

The salivary and blood MDA values were determined using the commercial OxiSelect MDA Adduct ELISA Kit (Cell Biolabs’ OxiSelect, San Diego, CA, USA) according to the manufacturer’s instructions. The kit had a sensitivity limit of 2 pmol/mg MDA adduct. All samples were tested in duplicate. MDA concentration was expressed in pmol/µL.

### 2.6. Treatment

Patients with periodontal disease were subjected to periodontal therapy comprising of scaling and root planing using Gracey curettes and ultrasonic scalers (Mini Piezon, Electro-Medical Systems, Nyon, Switzerland). The therapy was carried out in the form of 1–2 visits within 7 days without the use of antibiotics or antiseptics.

### 2.7. Follow-Up

Patients included in the experimental group underwent periodontal status assessment and had their saliva and blood samples taken during the first visit prior to commencing periodontal therapy, as well as three months after therapy completion, while the control group was subjected to the same procedure at the baseline only.

### 2.8. Statistical Analysis

Data collected as a part of the study were analyzed using the statistical package SPSS 16 for Windows. All values are presented as mean ± SD. Pearson χ2 test was used for testing relationships between individual pairs of observed attribute characteristics (gender). On the other hand, a *t*-test was performed to determine the differences in the mean values of numerical characteristics (age, number of teeth present, periodontal indices, comparison of index levels before and after the treatment, level of lipid peroxidation in the two groups, comparison of oxidative stress marker values obtained before and after treatment). The correlation between lipid peroxidation markers in saliva and blood was determined by Spearman’s rank correlation coefficient. We adopted the factorial ANOVA test to assess the influence of the confounding factors (sex, age and smoking status) on MDA values. The results for which the level of significance met the *p* < 0.05 criterion were interpreted as statistically significant.

## 3. Results

As can be seen from the flow chart of the experimental design presented in Figure 1, 87 individuals were initially eligible for participation in the study. However, eleven were excluded because they did not meet the inclusion criteria, while a further six refused to participate. After full-mouth periodontal clinical parameters examination, further 13 patients were excluded because they did not meet either Group A or Group B criteria. The remaining 57 patients were divided into Group A (37 patients) and Group B (20 patients). However, as seven patients from Group A did not return for the 3-month visit, their data were excluded from statistical analyses.

Therefore, the study sample comprised 50 patients, 30 of whom suffered from periodontitis (10 men and 20 women, average age 48.70 ± 9.68), while the remaining 20 had a healthy periodontium (9 men and 11 women, average age 46.25 ± 9.25).

The demographic characteristics of the patients are shown in Table 1. No statistically significant differences in age, gender and number of teeth present in the two groups were noted.

The mean values of the examined clinical parameters obtained before and after periodontal treatment are shown in Table 2. At baseline, the mean values of all periodontal indices were statistically significantly higher in the experimental group compared to the controls. Periodontal therapy led to a significant reduction in these indices in the group of patients with periodontitis.

The salivary level of the tested LPO marker (MDA) in the experimental group was significantly higher at baseline than in the control group, and its values were significantly reduced three months upon periodontal therapy completion (Table 2). Further, in patients with periodontal disease, markedly higher MDA values in plasma were noted relative to those measured in subjects with healthy periodontium (Table 2). Periodontal therapy, however, did not yield a statistically significant reduction in the MDA levels in plasma of patients with periodontal disease.

Patients with severe gingival inflammation had significantly higher MDA levels in saliva compared to patients with moderate gingival inflammation (Table 3). However, no statistically significant differences in the level of this marker in saliva were noted between the subgroups with PD ≥ 5 mm ≥ 20% and PD ≥ 5 mm < 20% (Table 4) or between those with CAL ≥ 3 mm and CAL < 3 mm (Table 5). The results obtained from plasma analysis indicate that the MDA level in patients with severe gingival inflammation was significantly higher compared to that in patients with moderate gingival inflammation (Table 3). When the patients were divided into subgroups based on the PD and CAL values, the differences noted between MDA values in plasma were found not to be statistically significant (Table 4 and Table 5).

At baseline, MDA levels in saliva showed a significant positive correlation with MDA levels in the plasma of patients with periodontal disease (r = 0.451, *p* = 0.012) (Figure 2).

The results yielded by the *t*-test and correlation analysis indicate the absence of statistically significant differences in the MDA levels in blood and saliva with respect to patients’ sex or age in either study group (Table 6).

Given the well-established prooxidative effect of tobacco smoke, the salivary and plasma MDA values in relation to the smoking status were analyzed in both study groups. In the group comprising patients with periodontitis, there were 10 smokers and 20 non-smokers, while in the control group, the ratio was 7 to 13 (Table 1). In smokers with periodontitis, the MDA value in saliva and plasma was 4.06 pmol/μL and 0.57 pmol/μL, respectively, while 2.46 pmol/μL and 0.46 pmol/μL were measured for non-smokers with periodontitis. In the control group, these values were 2.34 pmol/μL and 0.49 pmol/μL for smokers and 0.77 pmol/μL and 0.35 pmol/μL for non-smokers (Figure 3 and Figure 4).

The salivary and plasma MDA levels in smokers were statistically significantly higher than those measured for non-smokers in both groups. Therefore, a factorial ANOVA test was conducted to analyze the average salivary and plasma MDA levels according to study groups and smoking status. The obtained *p*-values suggest the presence of statistically significant differences in both salivary and plasma MDA between groups, as well as between smokers and non-smokers, but that there is no statistically significant interaction between groups or between smokers and non-smokers (Figure 3 and Figure 4).

## 4. Discussion

Misbalance between the production of free radicals and antioxidant protection leads to oxidative stress in the oral cavity. Existing studies show that the LPO terminal product concentrations in the gingival crevicular fluid and saliva of patients with periodontal disease are significantly higher compared to patients with healthy periodontium [16,18,32,33,34,35,36]. For example, Canakci et al. recorded an MDA concentration of 7.35 nmol/mL in the saliva of periodontitis patients, compared to 5.41 nmol/mL in subjects with healthy periodontium [13]. The results yielded by our investigation also indicate an increase in the salivary MDA levels in the presence of periodontal tissue inflammation. The value of this marker in patients with periodontal disease at baseline was 2.99 pmol/µL, which is significantly greater than 1.33 pmol/µL measured in healthy subjects. Higher MDA values obtained in the study conducted by Canakci et al. can be attributed to a greater degree of destruction to the periodontal supporting structures than was observed in our patients, as well as differences in methodology [13]. Specifically, this group of authors used stimulated saliva, whereas unstimulated saliva was utilized in our study. It is known that, during the sampling of stimulated saliva, greater quantities of gingival crevicular fluid are exerted into the saliva, which can significantly increase the values of the tested markers.

Empirical evidence supports the view that LPO marker levels can be used to estimate the extent of periodontium destruction [37]. This is confirmed by the findings reported by Khalili et al., who noted higher MDA values in patients with periodontal disease relative to healthy controls [11]. In addition, these authors found differences in the values of this marker in patients with mild, moderate and severe forms of the disease and reported a significant correlation between MDA and papilla bleeding index, probing depth and clinical attachment level. On the other hand, based on a comparison of the clinical parameters of periodontal disease with the MDA levels in saliva, Dakovic reported the presence of a link between the level of this marker and the degree of inflammation, but not periodontal probing depth [12]. As periodontitis is a cyclical process, whereby periods of remission alternate with periods of exacerbation, marked by the activation of all signs of inflammation, this author postulated that high MDA levels in patients with periodontal disease might be a sign of active processes in the periodontium. The results obtained in our study are consistent with the findings reported by Dakovic [12]. The salivary MDA levels in our patients were primarily influenced by gingival inflammation. Specifically, patients with severe gingival inflammation had significantly higher MDA values (3.62 pmol/µL) compared to patients with moderate inflammation (2.42 pmol/µL) (*p* = 0.045). The impact of probing depth, and clinical attachment level in particular, was not statistically significant.

In our study, periodontal therapy led to a significant reduction in the LPO levels in the saliva of patients with periodontal disease. Dakovic also reported a 58% reduction in the MDA levels in saliva of patients with the periodontal disease following treatment and concluded that, along with the elimination of gingival inflammation, the products of cell oxidative damage are also neutralized by periodontal therapy [12]. Based on the analysis of the salivary MDA values obtained after treatment in relation to the extent of gingival inflammation, probing depth and clinical attachment level, we can conclude that the reduction in MDA was primarily caused by the reduction in probing depth. Specifically, in the group of patients with greater PD, the difference in the MDA values before and after treatment was 1.06 pmol/μL, compared to only 0.53 pmol/μL measured for the group with lower PD (*p* < 0.05).

Presently, there is no consensus on the effect of periodontal disease on the occurrence of systemic oxidative stress. While some authors report higher levels of LPO markers in the blood of patients with periodontal disease relative to healthy controls [14,17,38], other studies reveal no significant differences [16,35,39,40]. The results yielded by our study show statistically significantly higher MDA levels in the blood of patients with periodontal disease compared to the levels measured in healthy subjects. Although it is possible that the increased MDA plasma concentration is due to some other pathological processes in the body, we nonetheless postulate that it is indeed the result of periodontal disease. This assertion is based on the fact that all our patients were systemically healthy and the presence of a statistically significant positive correlation between the MDA levels in saliva and in the blood (Figure 2).

Upon examining the process of LPO in patients with periodontal disease, Bastos et al. reported a significant correlation between the MDA levels in the blood and locally produced inflammatory cytokines (IL-10 and TNFα), highlighting that this marker is a valid indicator of the inflammatory process severity [41]. Our results also show that gingival inflammation was the primary factor in the high MDA value in the blood of patients with periodontal disease. Specifically, in patients with severe inflammation (0.55 pmol/µL), this marker was statistically significantly higher than in patients with moderate gingivitis (0.45 pmol/µL) (*p* = 0.047).

The periodontal therapy, however, did not result in a decline in the MDA concentration in the blood of our patients. Although the MDA value in the blood of our patients with periodontal disease at baseline was primarily affected by the extent of gingival inflammation, marked improvement in this condition after the therapy was not accompanied by a significant reduction in MDA. In fact, based on our findings, the decrease in MDA that was observed three months after therapy completion was mainly influenced by the reduction in probing depth in patients with the initially greater PD. This finding could point to the conclusion that periodontal therapy may have a greater benefit for patients with severe periodontal disease, as it results in a more pronounced reduction in circulating LPO. This conclusion is supported by the findings reported by Ambati et al. based on a sample of patients with higher PD values, in whom periodontal therapy resulted in statistically significantly lower serum MDA levels [42].

As a part of this research, we examined the influence of smoking on the level of lipid peroxidation in patients with and without periodontitis, given that smoking increases exposure to free radicals while also reducing the body’s antioxidant protection [43]. Moreover, smoking is one of the main contributing factors in the development of periodontitis, as it affects many aspects of the host’s immune response. The results yielded by our analyses indicate that the salivary and plasma MDA levels were much higher in smokers, irrespective of their periodontal status. These findings concur with the results published by other authors. For example, Guentsch et al. [44] reported a progressive increase in the salivary MDA values from healthy non-smokers to healthy smokers and further to non-smokers and smokers with periodontitis. Garg et al. [45] similarly noted that the MDA levels in gingival tissue are affected by the number of cigarettes smoked. Similar to our results, these authors recorded significantly higher MDA values in the blood of smokers compared to non-smokers with periodontitis.

We deliberately chose not to exclude smokers from our research, as our aim was to assess oxidative stress levels in the general population, and in our country, a significant proportion of adults smoke. For the same reason, we did not exclude obesity and physical activity as the risk factors for the emergence of oxidative stress. As the control and the experimental group were comparable and both included smokers, any influence of smoking on our findings would be minimized, as confirmed by the factorial ANOVA test results (Figure 3 and Figure 4).

## 5. Conclusions

Based on the results reported here, it can be concluded that salivary and plasma MDA (as a final LPO product) levels were significantly higher in patients with periodontal disease compared to healthy subjects. The level of this marker was most significantly impacted by the severity of gingival inflammation. However, only the salivary MDA levels were significantly reduced by periodontal therapy. Therefore, even though these findings are based on a relatively small sample, they point to the potential link between periodontitis and local and systemic oxidative stress, as well as highlight the benefits of periodontal therapy in mitigating this issue. Further research based on larger patient samples and examining a greater number of oxidative stress markers is nonetheless needed to confirm our results and expand our findings.

## Figures and Tables

**Figure 1 jcm-11-03617-f001:**
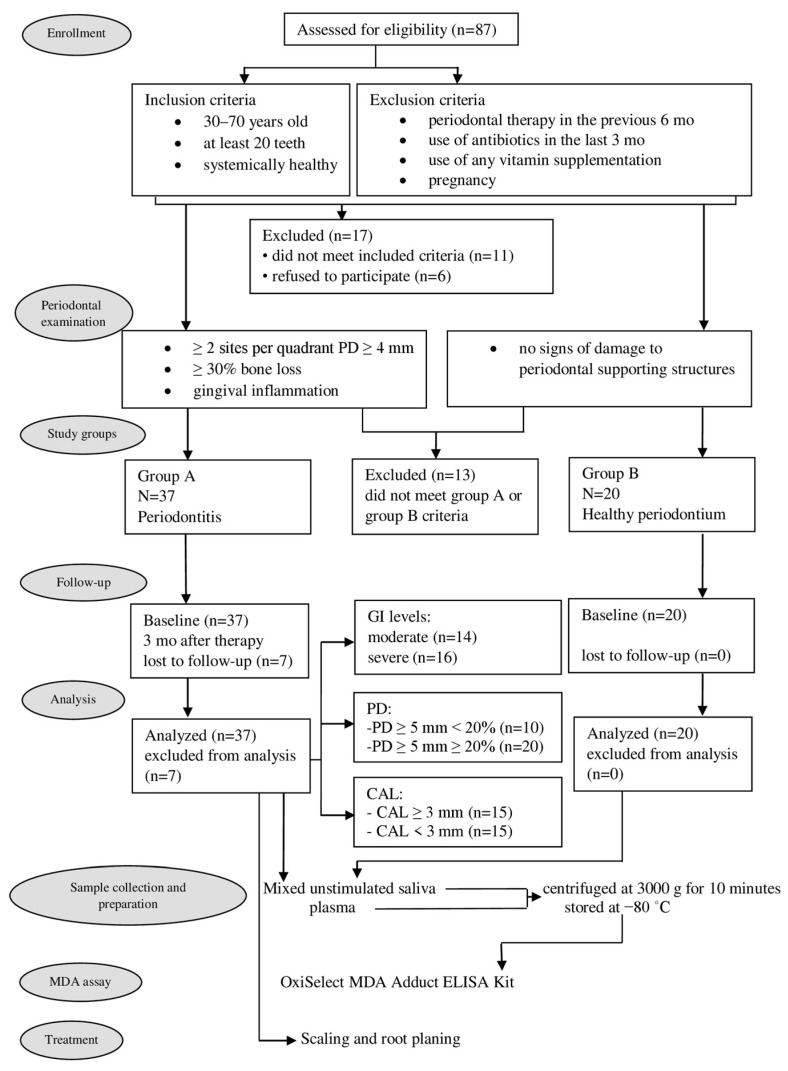
Flow chart of clinical study. GI—gingival index; PD—probing depth; CAL—clinical attachment level; MDA—malondialdehyde.

**Figure 2 jcm-11-03617-f002:**
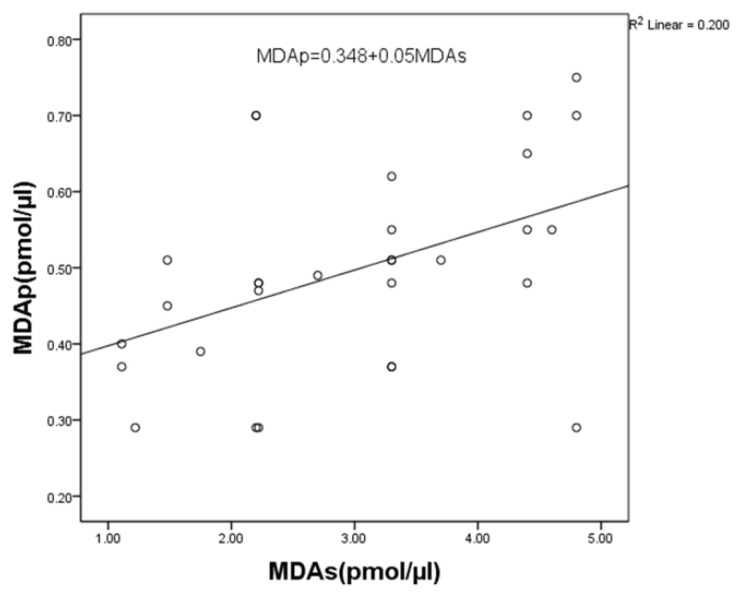
Correlation between MDA levels in saliva and plasma in periodontitis patients. Statistically significant at *p* < 0.05. MDAp—malondialdehyde in plasma; MDAs—malondialdehyde in saliva.

**Figure 3 jcm-11-03617-f003:**
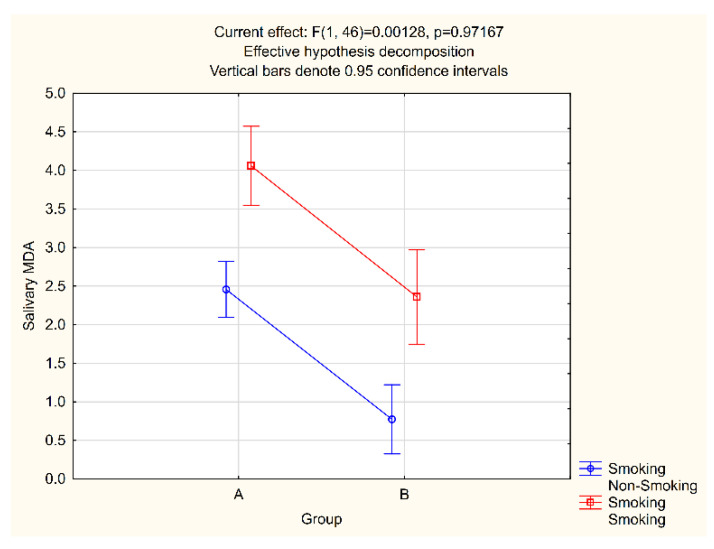
Salivary MDA and smoking interaction.

**Figure 4 jcm-11-03617-f004:**
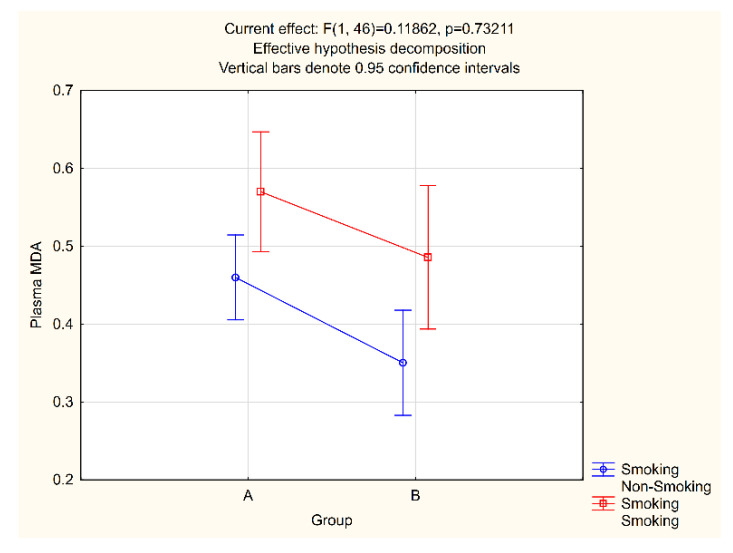
Plasma MDA and smoking interaction.

**Table 1 jcm-11-03617-t001:** Patients’ demographic characteristics.

	Group A (*n* = 30)	Group B (*n* = 20)	*p*-Value
Gender, *n* (%) Male Female	10 (33.3%) 20 (66.7%)	9 (45%) 11 (55%)	0.553
Age in years (mean ± SD)	48.70 ± 9.68	46.25 ± 9.25	0.472
Number of teeth (mean ± SD)	22.54 ± 2.14	25.34 ± 3.94	0.487
Smoking, *n* (%) Yes No	10 (33.3%) 20 (66.7%)	7 (35%) 13 (65%)	1.000

**Table 2 jcm-11-03617-t002:** Periodontal indices and MDA levels at baseline and three months upon therapy completion (mean ± SD, minimum−maximum).

	Group A Baseline	Group A Three Months after Therapy	Group B
PI	1.40 ± 0.43 (0.83−1.97)	0.35 ± 0.22 (0.08−1.12) ^a^	0.32 ± 0.28 (0.04−1.01) ^b^
GI	1.79 ± 0.64 (1.08−2.79)	0.25 ± 0.33 (0.02−1.31) ^a^	0.19 ± 0.37 (0.00−0.50) ^b^
PBI	1.57 ± 0.81 (0.64−3.47)	0.71 ± 0.43 (0.08−1.73) ^a^	0.28 ± 0.39 (0.00−1.78) ^b,c^
PD (mm)	3.14 ± 0.56 (2.87−4.42)	2.59 ± 0.45 (1.95−3.81) ^a^	1.45 ± 0.18 (1.13−1.71) ^b,c^
CAL (mm)	2.70 ± 1.03 (1.76−5.50)	2.19 ± 0.81 (0.59−4.08) ^a^	0.43 ± 0.56 (0.00−1.09) ^b,c^
MDA-saliva (pmol/μL)	2.99 ± 1.21 (1.11−4.80)	2.14 ± 0.95 (1.11−4.80) ^a^	1.33 ± 0.92 (0.23−3.70) ^b,c^
MDA-plasma (pmol/μL)	0.50 ± 0.13 (0.29−0.70)	0.47 ± 0.11 (0.29−0.70)	0.40 ± 0.13 (0.13−0.62) ^b^

^a^ Statistically significant difference compared with the baseline values (*p* < 0.05). ^b^ The difference between Group A and Group B was significant at the baseline (*p* < 0.05). ^c^ Statistically significant difference compared with the values after therapy (*p* < 0.05).

**Table 3 jcm-11-03617-t003:** Periodontal indices and MDA levels at baseline and three months upon therapy completion in groups with moderate and severe gingival inflammation (GI).

	GI Moderate (*n* = 14)	GI Severe (*n* = 16)
PI—baseline	1.08 ± 0.27	1.76 ± 0.11 ^a^
PI—3 mo after therapy	0.31 ± 0.10 ^b^	0.67 ± 0.35 ^b^
GI—baseline	1.37 ± 0.26	2.28 ± 0.21 ^a^
GI—3 mo after therapy	0.41 ± 0.20 ^b^	0.68 ± 0.52 ^b^
PBI—baseline	1.89 ± 0.39	2.01 ± 0.77 ^a^
PBI—3 mo after therapy	0.56 ± 0.27 ^b^	0.88 ± 0.53 ^b^
PD (mm)—baseline	2.88 ± 0.40	3.30 ± 0.57 ^a^
PD (mm)—3 mo after therapy	2.40 ± 0.31 ^b^	2.81 ± 0.49 ^b^
CAL (mm)—baseline	2.48 ± 0.78	2.96 ± 1.32
CAL (mm)—3 mo after therapy	1.69 ± 0.73 ^b^	1.91 ± 1.03 ^b^
MDA-saliva (pmol/μL)—baseline	2.42 ± 1.10	3.62 ± 1.22 ^a^
MDA-saliva (pmol/μL)—3 mo after therapy	1.83 ± 0.73 ^b^	2.50 ± 1.07 ^b^
MDA-plasma (pmol/μL)—baseline	0.45 ± 0.12	0.55 ± 0.13 ^a^
MDA-plasma (pmol/μL)—3 mo after therapy	0.43 ± 0.12	0.51 ± 0.09

^a^ The difference between subgroups with moderate and severe inflammation was significant at baseline (*p* < 0.05). ^b^ Statistically significant difference compared with the baseline values (*p* < 0.05).

**Table 4 jcm-11-03617-t004:** Periodontal indices and MDA levels at baseline and three months upon therapy completion in groups with PD ≥ 5 mm < 20% and PD ≥ 5 mm ≥ 20%.

	PD ≥ 5 mm < 20% (*n* = 10)	PD ≥ 5 mm ≥ 20% (*n* = 20)
PI—baseline	1.24 ± 0.32	1.49 ± 0.43
PI—3 mo after therapy	0.37 ± 0.23 ^b^	0.54 ± 0.33 ^b^
GI—baseline	1.55 ± 0.54	1.93 ± 0.45
GI—3 mo after therapy	0.48 ± 0.34 ^b^	0.59 ± 0.44 ^b^
PBI—baseline	1.41 ± 0.74	1.67 ± 0.71
PBI—3 mo after therapy	0.59 ± 0.40 ^b^	0.78 ± 0.45 ^b^
PD (mm)—baseline	2.58 ± 0.21	3.46 ± 0.41 ^a^
PD (mm)—3 mo after therapy	2.21 ± 0.21 ^b^	2.81 ± 0.40 ^b^
CAL (mm)—baseline	2.22 ± 0.91	2.99 ± 1.01 ^a^
CAL (mm)—3 mo after therapy	1.45 ± 0.75 ^b^	1.98 ± 0.90 ^b^
MDA-saliva (pmol/μL)—baseline	2.62 ± 1.11	3.21 ± 1.23
MDA-saliva (pmol/μL)—3 mo after therapy	2.09 ± 0.88 ^b^	2.15 ± 1.03 ^b^
MDA-plasma (pmol/μL)—baseline	0.44 ± 0.12	0.53 ± 0.12
MDA-plasma (pmol/μL)—3 mo after therapy	0.42 ± 0.12	0.45 ± 0.11 ^b^

^a^ The difference between subgroups with PD ≥ 5 mm < 20% and PD ≥ 5 mm ≥ 20% was significant at baseline (*p* < 0.05). ^b^ Statistically significant difference compared with the baseline values (*p* < 0.05).

**Table 5 jcm-11-03617-t005:** Periodontal indices and MDA levels at baseline and three months upon therapy completion in groups with CAL < 3 mm and CAL ≥ 3 mm.

	CAL < 3 mm (*n* = 15)	CAL ≥ 3 mm (*n* = 15)
PI—baseline	1.22 ± 0.37	1.58 ± 0.36 ^a^
PI—3 mo after therapy	0.43 ± 0.28 ^b^	0.53 ± 0.33 ^b^
GI—baseline	1.65 ± 0.50	1,94 ± 0.43
GI—3 mo after therapy	0.49 ± 0.34 ^b^	0.60 ± 0.46 ^b^
PBI—baseline	1.38 ± 0.78	1.77 ± 0.62
PBI—3 mo after therapy	0.53 ± 0.33 ^b^	0.88 ± 0.46 ^b^
PD (mm)—baseline	3.05 ± 0.61	3.23 ± 0.50
PD (mm)—3 mo after therapy	2.42 ± 0.31 ^b^	2.75 ± 0.52 ^b^
CAL (mm)—baseline	1.99 ± 0.72	3.42 ± 0.77 ^a^
CAL (mm)—3 mo after therapy	1.35 ± 0.69 ^b^	2.23 ± 0.83 ^b^
MDA-saliva (pmol/μL)—baseline	2.80 ± 0.91	3.18 ± 1.45
MDA-saliva (pmol/μL)—3 mo after therapy	2.09 ± 0.74 ^b^	2.18 ± 1.15 ^b^
MDA-plasma (pmol/μL)—baseline	0.46 ± 0.13	0.54 ± 0.13
MDA-plasma (pmol/μL)—3 mo after therapy	0.44 ± 0.08	0.49 ± 0.14

^a^ The difference between subgroups with CAL < 3 mm and CAL ≥ 3 mm was significant at baseline (*p* < 0.05). ^b^ Statistically significant difference compared with the baseline values (*p* < 0.05).

**Table 6 jcm-11-03617-t006:** The influence of sex, age and smoking status on the salivary and plasma MDA levels.

Parameter	MDA-Plasma		MDA-Saliva	
Variable	Statistics	*p*-Value	Statistics	*p*-Value
Group	2.593 ^a^	0.013	5.226 ^a^	<0.001
Smoking	−3.077 ^a^	0.003	−4.543 ^a^	<0.001
Gender	0.073 ^a^	0.942	−1.249 ^a^	0.218
Age	0.155 ^b^	0.283	0.014 ^b^	0.926

^a^ Independent *t*-test, ^b^ Correlation coefficient.

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
