# Peer review of "Lipid Peroxidation Levels in Saliva and Plasma of Patients Suffering from Periodontitis"

_jcm, 2022, doi:10.3390/jcm11133617_

Round 1

Reviewer 1 Report

The topic is interesting for a worldwide medical and dentist. This is an interesting work. However, to my understanding, some major methodological and statistical issues could bias and not address the objective.

1 How to determine the sample size?

2 It should give some reason of setting the age of included subjects were 30-70 years old.

3 Periodontal status should be defined by the 2018 classification scheme in the group of patients with periodontal disease.

4 Gingival crevicular fluid samples would be more likely to reflect periodontal inflammation and damage to periodontal tissues than salivary samples.

5 In view of the possible impact of smoking, study subjects who smoked cigarettes should be excluded. 

6 Multi-factor-analysis after adjusting the confounding factors should be made to shows the real association between MDA levels in saliva and plasma in periodontitis patients.

7 The difference between two subgroups should be added.

Reviewer 2 Report

Dear authors,

Below are my comments regarding your submitted article. 

- I would recommend reorganizing the abstract to be more concise. 

- I think that at the end of the introduction section would be appropriate to be specified the aim of the study, not only the objectives.

- I would suggest the rephrasing of the conclusion to be more eloquent.

- I would also suggest reconsidering the Reference section since some cited articles are quite old (2001, 2002).

This manuscript is a resubmission of an earlier submission. The following is a list of the peer review reports and author responses from that submission.